# Characteristics of Si (C,N) Silicon Carbonitride Layers on the Surface of Ni–Cr Alloys Used in Dental Prosthetics

**DOI:** 10.3390/ma17102450

**Published:** 2024-05-19

**Authors:** Leszek Klimek, Marcin Makówka, Anna Sobczyk-Guzenda, Zofia Kula

**Affiliations:** 1Institute of Materials Science and Engineering, Faculty of Mechanical Engineering, Lodz University of Technology, B. Stefanowskiego 1/15, 90-924 Lodz, Poland; leszek.klimek@p.lodz.pl (L.K.); marcin.makowka@p.lodz.pl (M.M.); anna.sobczyk-guzenda@p.lodz.pl (A.S.-G.); 2Department of Dental Technology, Medical University of Lodz, Pomorska Str. 251, 92-213 Lodz, Poland

**Keywords:** dental alloys, Si (C,N) coatings, PVD coatings, texture

## Abstract

Chromium- and cobalt-based alloys, as well as chrome–nickel steels, are most used in dental prosthetics. Unfortunately, these alloys, especially nickel-based alloys, can cause allergic reactions. A disadvantage of these alloys is also insufficient corrosion resistance. To improve the properties of these alloys, amorphous Si (C,N) coatings were deposited on the surfaces of metal specimens. This paper characterizes coatings of silicon carbide nitrides, deposited by the magnetron sputtering method on the surface of nickel–chromium alloys used in dental prosthetics. Depending on the deposition parameters, coatings with varying carbon to nitrogen ratios were obtained. The study analyzed their structure and chemical and phase composition. In addition, a study of surface wettability and surface roughness was performed. Based on the results obtained, it was found that amorphous coatings of Si (C,N) type with thicknesses of 2 to 4.5 µm were obtained. All obtained coatings increase the value of surface free energy. The study showed that Si (C,N)-type films can be used in dental prosthetics as protective coatings.

## 1. Introduction

Metals and their alloys are materials that have been used in dental prosthodontics for a very long time in the production of fixed and removable restorations. The most common are alloys that contain chromium, nickel, and cobalt in their composition. Unfortunately, they can negatively affect the human body, causing allergic reactions [1,2,3]. Allergies to these alloys can appear not only on the oral mucosa, but also on the skin of the hands, feet, and the entire body [3,4,5]. They are caused by products of reactions released in corrosion processes of these alloys in the environment of the human body. Thus, the released metal ions penetrate the surrounding tissues, causing various types of allergic reactions. Therefore, the materials used in production of elements which may have prolonged direct contact with tissues of the human body must, among other things, have a high resistance to corrosion [6,7,8]. Reduced toxicity is possible with a coating barrier, which involves reducing the amount of substrate ions released into the surrounding environment. It should be remembered that electric tools are overly aggressive towards prosthetic supplementary elements made of metal alloys [8,9]. Even though base metal alloys have a low corrosion resistance compared to precious metals, they are still widely used in dental prosthetics and also orthodontics. The reason for this is that other properties of alloys are also required, for example high strength or durability [9] and economic considerations.

In 1990, the IARC (International Agency for Research on Cancer) recognized nickel as a factor that may cause cancer in humans [10,11,12]. In recent years, Cr and Co have also been classified as potentially carcinogenic and mutagenic materials [10]. Regulation (EU) 2017/745 reviewed all classification rules and established new guidelines that apply to the medical device market [12]. These standards apply to producers, researchers, doctors, and dentists. These regulations provide better protection and safety for patients. One regulation indicates that any medical device containing Co (usually in the form of an alloy in dentistry) complies with the regulation if the Co concentration is less than 0.1% (m/m) [10,12]. Therefore, medical manufacturers are obliged to present alternative solutions by May 2025. Therefore, it is justified to undertake as a research topic.

To reduce corrosion and thus the risk of adverse body reactions, various modifications are applied to the surface of parts made of these alloys. The most common modification is the deposition of coatings by means of PVD, CVD, and sol-gel methods, as well as the structuring of surface layers by laser ablation [13,14,15]. Many studies have shown that the mentioned modifications can significantly increase corrosion resistance [13,14,15,16,17,18]. As a result, the amount of metal ions released into the oral environment causing various allergies is reduced. Among the most common material solutions in the modification of dental alloys in the form of coatings are ceramic materials such as carbides, oxides, and transition metal nitrides [19,20,21,22], mainly due to their high strength and corrosion resistance. The current literature indicates that the use of such coatings reduces the rate of nickel ion transfer from the alloy into the patient’s body [23,24].

When considering the in-service working conditions of prosthetic components, it is also important to improve their tribological properties, such as resistance to abrasive wear, fretting, and others [25]. In addition, modification of the alloy surface with the proposed solution allows the reduced formation of bacterial biofilm [26,27]. This is very important, especially when treating missing teeth with prosthetic components in dental implants. The main reasons for the failure of this type of treatment include bacterial infections. Bacteria can accumulate around the implant abutment and contribute to the development of inflammation [28].

Despite many research works and achievements in the field of coatings, better solutions are still being sought. All researchers’ activities are focused on obtaining one thin coating with exceptionally good mechanical, functional, and biological properties (biological reaction or bactericidal properties). So far, all research has been based on the use of other elements or a different substrate. Most research focuses primarily on nitrides themselves or oxides such as TiO_2_ [29,30]. To date, most of the research work has involved studies on the modification of Ni-Cr alloys coated with Ti (C,N) [22,23,24]. A second important solution is a silicon carbide (SiC) coating, which has found application in the dental industry in the modification of dental implant surfaces. These coatings have gained recognition due to their very high hardness, as well as wear resistance. Further advantages of these materials are high biocompatibility and the absence of harmful side effects when in direct contact with living tissue [30]. Xinyi Xia and co-authors deposited SiCN coatings by means of the PECVD method, where the gas flow rates of ammonia (NH_3_), methane (CH_4_), and silane (SiH_4_) were precisely defined [29,30]. The introduction of nitrogen into silicon carbide provides antibacterial properties [29,30,31]. Travlou et al. [32] have shown that nitrogen-containing compounds can inhibit bacterial growth on the surface of alloys.

Our pioneering research presents the potential of application of SiCN coatings in dental prosthetics, deposited by means of the reactive magnetron sputtering (RMS) method. The literature indicates that so far, the main techniques used for the deposition of this type of coating have been ion-beam-assisted sputtering, high-power pulsed magnetron sputtering (HiPIMS)), thermal chemical vapor deposition (CVD), and plasma-enhanced deposition (PECVD) [29,30,31,32,33]. So far, Si (C,N) coatings have not been used for the modification of Ni-Cr alloy properties in dental prosthetics. The purpose of this work was to characterize silicon carbide coatings, deposited by the magnetron sputtering method on the surface of nickel–chromium alloys used in dental prosthetics. The study analyzed the chemical and phase composition on cross-sections of silicon carbide coatings. In addition, a study of wettability and surface roughness was performed.

## 2. Materials and Methods

The test materials were disks made of the Ni-Cr alloy Heraenium NA by Heraeus Kulzer (Hanau, Germany), 8 mm in diameter and 10 mm high. The samples were divided into five groups (five samples per group) denoted as No. 1 to No. 5, to cover them with coatings with different carbon and nitrogen contents, with the sixth group constituted by uncoated samples (No. 0).

The initial composition of the alloy determined by the X-ray fluorescence analysis technique with the use of a spectrometer SRS300 by SIEMENS (Berlin, Germany) is given in Table 1.

The samples were washed in a mixture of water and detergent and then in acetone in an ultrasonic cleaner before being mounted on the rotating workstation holder for coating deposition. They were then dried with a stream of compressed air and mounted on the workstation holder with care not to contaminate the faces of the coated samples. In addition, two monocrystalline Si (p-type) control samples measuring 10 × 10 × 0.525 mm were also mounted before each process.

The coatings were deposited by magnetron sputtering on a B.90 stand. The apparatus is equipped with four independent circular magnetrons with target diameters of 100 mm, connected to medium frequency power supplies. The magnetrons are arranged in such a geometrical configuration that the axes of their targets intersect near the axis of the sample holder. The holder allows the charge mounted on it to rotate in the vertical axis at speeds ranging from 0 to 12 rpm. In addition, this holder is isolated from the chamber walls and it is possible to connect an external power source to it. This source, in the form of a glow discharge power supply, can be used to clean the charge. During the process, it is also possible to polarize the charge by applying a potential from an external DC or AC power supply [34,35].

Four 107 × 10 mm pure Si (5N) targets were used in all coating deposition processes. Pure gases like acetylene (2.5N) and nitrogen (5N) were used as carbon and nitrogen sources during the processes. The working gas was pure argon (5N). The parameters of the five technological processes were chosen to obtain coatings with chemical compositions ranging from pure silicon carbide to pure silicon nitride, with the proportion of carbon to nitrogen varying proportionally. Additional working gases were introduced into the vacuum chamber from the fifth minute of the process.

The exact parameters for the deposition of a coating made of silicon carbides, silicon nitrides, and silicon carbonitrides are summarized in Table 2.

### 2.1. Chemical Composition

The chemical composition of the test sample was estimated using a JEOL JSM-6610LV (Tokyo, Japan) SEM (Scanning Electron Microscope) with an Oxford Instruments X-MAX 80 (Abingdon, Oxfordshire, England) module installed for energy dispersive X-ray (EDS) measurements using an accelerating voltage of 5 kV. Measurements were performed on pure Si samples, with a low accelerating voltage applied to minimize the depth from which the characteristic radiation originated, which was then used to measure qualitative and quantitative chemical composition.

Due to the limitations of the EDS method (in particular, the inability to accurately quantify light and heavy elements simultaneously, as well as to accurately analyze the carbon content and the possibility of carbon contamination of both the walls of the vacuum chamber of the microscope and the surface of the samples), the results of the proportion of carbon relative to the other elements can be subject to large error. To minimize errors, an analysis of the relative ratio of carbon to silicon was carried out, comparing the results obtained with those obtained for reference samples of SiN coatings, for which the measured carbon content was considered to be the contamination standard noise. All coatings in terms of their chemical composition were measured during a single test, to obtain the same carbon contamination conditions. For each group of coatings, two samples of monocrystalline silicon with deposited coatings were examined by SEM-EDS [36,37].

### 2.2. Phase Composition

In order to identify the phases and their crystallographic structure, X-ray diffraction studies were performed on flat coated surfaces of Ni alloy samples. They were carried out on a PANalytical Data Collector 4.0a diffractometer (Malvern, Worcestershire, United Kingdom). Visualization of the obtained data was performed with the PANalytical Data Viewer 1.4 software. A Cu Kα X-ray tube with a radiation wavelength kα = 1.5406 Å and an accelerating voltage of 45 kV, with a current of 40 mA, was used. Low-angle diffraction was performed with an incidence angle of the primary beam of 1°. A range of angles 2Θ from 30 to 80° were analyzed with a step equal to 0.05° and a step time of 2 s. On the basis of the obtained diffraction spectra, the phases present in Si (C,N)-type coatings were identified by comparing the obtained spectra with the patterns provided in the crystallographic database ACDD PDF 4. The results are shown in for coatings from pure SiC, through to SiCN, to pure SiN, respectively. One Ni-Cr alloy sample from each group with a different coating was selected for this investigation.

### 2.3. Cross-Section of Coatings and Their Thicknesses

The thicknesses of the coatings were determined on a brittle cross-section of the coatings, obtained after breaking silicon control samples coated with the coatings. Imaging was performed by the SEM technique on a JEOL JSM-6610LV bench (Tokyo, Japan) operating in secondary electron (SE) mode using an accelerating voltage of 20 kV. For each group of specimens, two samples of monocrystalline silicon with deposited coatings were examined by the SEM-EDS method. At least three coating thickness measurements were taken from different points of their cross-sections to determine the mean value of thickness of the coatings [36,37].

### 2.4. Wettability and Surface Free Energy (SFE)

We prepared three samples from each group for this test. The wettability of the coatings was measured using the sessile drop technique with two liquids of different polarity and known surface tension: distilled water and diiodomethane (Sigma Aldrich, Saint Louis, MO, USA). Measurements were made using the FM40 Easy Drop system with Drop Shape Analysis software V1.72-02 (Krüss GmbH, Hamburg, Germany). Wetting measurements were carried out in an ambient environment (measurement temperature was 23 ± 1 °C and air humidity was 50 ± 5%). The study used the sessile drop method (volume 0.8 µL). Measurements were repeated in at least 7 different locations. The wettability measurements were followed by the assessment of the surface free energy (SFE). This was based on the Owens–Wendt theoretical model. The Owens–Wendt method involves determining the dispersive and polar components of the SFE based on the Bethelot hypothesis, which states that the interactions between the molecules of two substances located in their surface layer are equal to the geometric mean intermolecular interactions within each substance. To determine the SFE, two measured liquids (water—polar; diiodomethane—dispersive) must be used, whose surface tension and polar and dispersive components are known. Distilled water is a highly polar liquid because its polar component is 51 mN/m and the dispersive component is 21.8 mN/m. The components of diiodomethane SFE are as follows: polar—2.4 mN/m; dispersive—48.6 mN/m [38,39].
(1)$$\gamma_{L}=\gamma_{L}^{d}+\gamma_{L}^{p}$$
(2)$$\gamma_{S}=\gamma_{S}^{d}+\gamma_{S}^{p}$$
(3)$$1+\text{cos}\Theta =\frac{2}{\gamma_{L}}\left[{\left(\gamma_{L}^{d}\gamma_{S}^{d}\right)}^{\frac{1}{2}}+{\left(\gamma_{L}^{p}\gamma_{S}^{p}\right)}^{\frac{1}{2}}\right]$$
where: *γ_L_* is the free surface energy of the liquid in equilibrium with the saturated water vapor of the liquid, *γ_S_* is the free surface energy of the solid in equilibrium with the saturated vapor, *γ^d^_L_*, *γ^p^_L_* are the dispersive and polar components of the measuring liquids’ surface energy, and *γ^d^_S_*, *γ^p^_S_* are the dispersive and polar components of SFE of the examined materials [40].

### 2.5. Surface Roughness

The roughness test was carried out using a Mitutoyo profilometer (Sakado, Japan). The sample was mounted on a holder and then set perpendicular to the profilometer needle. We prepared three samples from each group for this test. Measurements were made in three different directions of the moving needle along the sample surface. The obtained results were exported to a computer, which then calculated the roughness parameters. The test was performed based on the PN-ISO 4288:2011 standard. Parameters: Ra (arithmetical mean deviation of the assessed profile); Rz (maximum height of the profile); Rp (maximum peak height; Rq (root mean squared); and RV0 (retention volume). The measured profile presents the total effects of roughness, waviness, and shape errors. The calculation parameters are presented in the table below (Table 3). [41].

## 3. Results and Discussion

### 3.1. Chemical Composition

The studies performed using the SEM-EDS technique made it possible to determine the changes in the proportions of elements such as Si, C, and N in the coatings. Table 4 collects the results of the chemical composition measurements of all the produced coatings. Figure 1 shows an example of an EDS spectra of the investigated SiCN coating (process no. 3), while Figure 2 shows the changes in the ratios of carbon to nitrogen shares along with the change in the ratio of methane to nitrogen flow. The analysis was carried out on selected areas of the silicon samples, but on an area of no less than 0.5 mm^2^, in order to obtain better statistics for the results.

Based on the results of the EDS analysis, it was found that the composition of the produced coatings contains Si, N and C, as well as O. The proportion of N and C changes proportionally to the change in the ratio of C_2_H_2_ to N_2_ flow. For the C_2_H_2_ flow alone, a coating containing only Si and C was obtained, while for the N_2_ flow alone, the coating composition included Si and N. In all cases, the proportion of oxygen was up to a maximum of 2 at. %. The dependence of changes in N and C shares shows a linear relationship with respect to the flow of their precursors.

### 3.2. Phase Composition

Figure 3 shows diffractograms of the studied coatings on Ni alloy substrates, highlighting the identified reflections and the corresponding patterns from the ICDD database. These diffractograms were processed in XRD result analysis software before further processing, i.e., background and noise were removed.

Analysis of XRD diffraction patterns of coatings on Ni alloy substrates revealed the diffraction maxima originating from Ni solution (ICDD reference: 01-087-3773). Following peaks were identified: Ni (111) at the 2θ angle of 43.7 deg., Ni (200) at 50.9 deg., and Ni (220) at 74.9 deg. These reflections come from the substrate and no reflections were found from other phases, including phases typical of Si, SiN and SiC, SiCN. What is noticeable, in the wide range from about 20 deg. to 40 deg. of 2θ, is that a fuzzy reflection of low intensity was observed. The origin of this phenomenon is twofold: it may indicate the amorphous nature of the examined material, but it can also come from the scattering of the incident at low angles of the X-ray beam (for the method used, the angle of incident of the beam is 1 deg.), observed precisely when examining the reflected beam at low angles such as 2θ of 20–30 deg. No significant shift in peaks or change in their width originating from the phases present in the substrate was observed. Noticeable is a change in their relative intensity. However, this phenomenon may be due to the crystallographic orientation and texture of the alloy grains near the surface in the substrate in the investigated volume for a given sample, rather than to the effect of the coating itself.

The second crucial factor to consider is the depth of X-ray penetration into the sample and the depth from the surface from which we obtain information regarding phase composition and crystallinity. In the Bragg–Brentano system, where the angle of incidence is equal to the angle of reflection, the depth from which we obtain information is calculated in tens or even hundreds of micrometers. This relationship is closely related to the angle of incidence and it is stated that small entry angles lead to a smaller information depth than large entry angles [42]. In the case of the samples studied, a fixed angle of incidence equal to 1 deg. was used, which in turn translated into much smaller depths from which information was obtained (like single micrometers) regarding phase composition and crystallinity. Considering the absence of reflections from phases typical of the Si with C and N system, the low incidence angle of the radiation beam and coating thicknesses of around 2.4 to 4.2 µm, it can be concluded that the coating material is amorphous.

### 3.3. Coatings Thickness

Figure 4 shows the coating cross-sections with their thicknesses, while Figure 5 shows the change in coating thicknesses as a function of the composition of the working atmosphere.

The measured coating thicknesses on brittle coating cross-sections range from 2.6 to 4.2 µm. They vary depending on the deposition conditions: more specifically, the ratio of C_2_H_2_ and N_2_ reactive gas flows.

Considering the similar densities of SiC and SiN, as well as the obtained dependencies of coating thicknesses on reactive gas flows, it follows that the deposition rate of the SiN coating is around 30% higher than that of the SiC coating. This may be because during the deposition process the targets of pure Si are contaminated by the reaction products of the Si surface with reactive gases. The SiC compounds formed on the target’s surface, which are more difficult to sputter, cause the coating deposition process to slow down.

Cross-sectional observations of the coatings deposited on the silicon substrate allowed the determination of compactness and defects in the coating. No cracks or chipping of the coating was found by analyzing the cross-section at multiple points. The coating is compact and tight, which in turn is a critical condition for it to be a diffusion barrier both against the access of body fluids directly into the implant material, as well as exchange: the influx of ions from the direction of the implant material into the soft tissue.

### 3.4. Wettability and Surface Free Energy (SFE)

Table 5 shows the values of the measured contact angles for substrates with different coatings (No. 1–No. 5). The value of the water contact angle for the surface of a clean substrate is 81.5 deg. For the tested coatings, the value of this angle varies in a range from 65.4 to 86.8 deg. The obtained results indicate that the water contact angle values for most coatings are close to the boundary distinguishing hydrophilic from hydrophobic surfaces. Lower values of contact angles compared to the clean substrate were obtained for non-SiCN coatings. Knowing the nature of the surface (wettability) is very important in the case of materials that will come into contact with living tissues in the future. Wettability determines protein adsorption, blood coagulation, and biological response. Protein adsorption is the first process that occurs when a biomaterial comes into contact with blood and is the result of the interaction between the surface and the solvent or is a result of intermolecular interactions (e.g., van der Waals forces and hydrogen bonds) between the protein and the surface. Most used biomaterials are hydrophobic and have a high affinity for many proteins. Immediately after implantation, the biomaterial is covered with a layer of proteins. These are albumins, fibrinogen, fibronectin, immunoglobulins, and von Willebrand factors. Through hydrophobic interactions, these proteins, due to their different conformations and the presence of hydrophobic domains, adhere to hydrophobic surfaces [43,44]. This means that in this respect, SiCN coatings are the most preferential, especially coating No. 2.

Table 6 presents the values of surface free energy and its dispersive and polar components. The polar component SFE is the sum of the forces of hydrogen, acid–base, and inductive interactions, while the dispersion component determines the size of intermolecular interactions called London forces. Dispersion interactions always occur, while polar interactions occur only when polarization of chemical bonds takes place. The lowest value of the polar component was obtained for SiCN coating No. 2, it is 2.9 mJ/m^2^, and the highest value was obtained for SiN coating No. 5. The value of surface free energy for a clean substrate is 28 mJ/m^2^. Each modification of the substrate increases the value of surface free energy, which ranges from 33.7 to 40.2 mJ/m^2^. From the point of view of the biocompatibility of the material, this is a desirable phenomenon, because the area between 20 and 40 mJ/m^2^ is the hypothetical region of biocompatibility. However, materials with a critical surface free energy value above 40 mJ/m^2^ are particularly desirable in bone applications. For this reason, coating No. 5—SiN—seems to be the best. In turn, in dentistry and prosthetics, high values of surface free energy may lead to increased susceptibility to dental plaque, the formation of which is initiated by the adhesion of bacteria found in the oral cavity. The higher the surface free energy, the easier it is for bacteria to adhere to these surfaces [45], which may at the same time exclude SiN coating No. 5 in favor of SiCN coatings.

### 3.5. Surface Roughness

The surface roughness measurement results are presented in Table 7 and Figure 6 and Figure 7. Samples were divided into six official groups. Measurements were made in three different software applications to validate the test results. The roughness of each calculation is based on the arithmetic mean of three results (µm). However, the basic parameters of Ra and Rz are mostly used to describe the unevenness. In this work, additional roughness parameters such as Rq, Rp and Rv are used. Table 7 summarizes the roughness calculations for the following parameters: Ra, Rq, Rz, Rp, and Rv.

For the roughness parameters shown in Table 7, the results decrease. In the case of sample No. 0, the highest values are observed (Ra = 0.29 µm). No. 0 is a sample where there is no coating on the substrate. Sample No. 5 and No. 2 have the lowest roughness parameters. These are the parameters (Ra = 0.44 µm) for sample No. 5 and (Ra = 0.055 µm) for sample No. 2. Average Ra and Rz values (µm) of groups under research study were much lower compared to group 0.

Figure 6 present graphs for an uncoated alloy substrate. Figure 7 present graphs for substrate layers of deposited of Si (C,N) silicon carbonitride. In Figure 6 and Figure 7, differences between the prepared profiles are visible. Differences can be observed in the width of roughness unevenness.

The tests of surface roughness enabled determining the coating’s impact on changes in sample roughness. The application of coatings completely changed the surface of the alloy. The coatings used on the Ni-Cr metal surface are characterized by a smoother surface compared to surfaces without silicon carbonitride layers. The observed differences may be important in medical and dental applications [46,47,48,49,50].

To achieve implant success, the following factors must be taken into account: implant design, implant material, implant surface, and bone quality. In assessing the osseointegration properties of surfaces, topography, chemistry, charge, and wettability are important [50,51]. Bacterial adhesion is influenced by the roughness of the implant surface and its chemical and physical properties, which is why it is so important to maintain this parameter in the context of dental and medical implants [51,52,53]. Surfaces with a developed, rough surface improve bone implant contact, but are more susceptible to biofilm accumulation and are more difficult to clean. Patients with a dental implant are at risk of peri-implant disease, which can be defined as a bacterial infection that causes inflammation of the soft tissues around dental implants [51]. The bacteria causing periodontal diseases are *Porphyromonas gingivalis* and *Treponema denticola* [54,55]. Microorganisms associated with the implant surface in the presence of healthy tissues around the implant are *Gram-positive* cocci and rods [56]. Although it is more difficult for bacteria to settle on smooth surfaces, efforts should be made to obtain a surface with antibacterial properties. There are findings in the literature attributed to authors Freire et al. [57] that an endogenous species (*A. actino-mycetencomitans strain*) is able to induce biofilm formation on smooth and rough surfaces in vitro.

To sum up, the production of Si (C,N) coatings on the surface of a Ni-Cr alloy for dental applications is a promising solution. Assessment of coated surfaces indicates smooth surfaces, which is important for dental applications. The resulting silicon carbonitride layers can create a protective barrier between the metal implant and human bone or tissue, thereby eliminating allergic reactions caused by soluble metal ions (such as Co, Cr, and especially Ni) in patients. Based on the publication by Chiang I and co-authors. [57], Si (C,N) layers have antibacterial properties against *S. mutans*. Contact angle measurements showed that they exhibited higher contact angles compared to untreated silicon surfaces, indicating increased hydrophobicity.

## 4. Conclusions

The main characteristics of obtained Si (C,N)-type coatings is their homogenous, flawless structure. Depending on application parameters, coatings of diverse ratios of carbon and nitrogen were obtained. Supposedly, this should have major influence on their mechanical properties, corrosion resistance, and operational properties in the oral cavity’s environment. In all the coatings, the contact angle was below 90°. Supposedly, this should reduce biofilm formation on coated prosthetic elements. Use of those coatings reduces surface roughness and facilities hygiene. Based on the conducted research, examined Si (C,N)-type coatings can be used in prosthodontics as protective coatings for base metal alloys. These properties can be precisely modulated through a variety of strategies, including the incorporation of doping elements, nitrogen flow rate control, temperature control, and specific post-processing techniques. The obtained coatings are characterized by a uniform structure without damage. This means that they can be used successfully in dental prosthetics.

The authors plan to continue further research in order to determine the mechanical, biological, and corrosion resistance properties of silicon carbide nitride coatings on the surface of nickel–chromium alloys used in prosthodontics.

## Figures and Tables

**Figure 1 materials-17-02450-f001:**
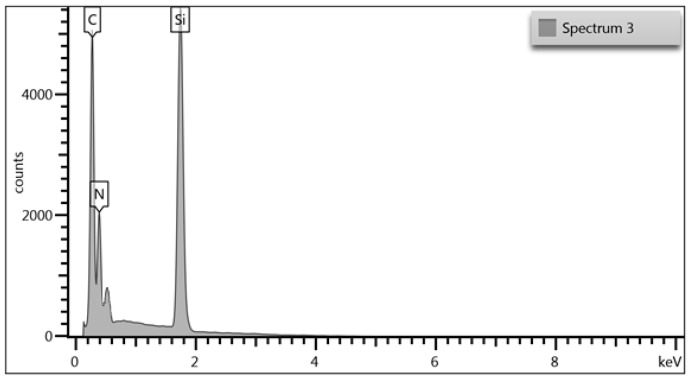
EDS spectra example for SiCN coating deposited in process no. 3.

**Figure 2 materials-17-02450-f002:**
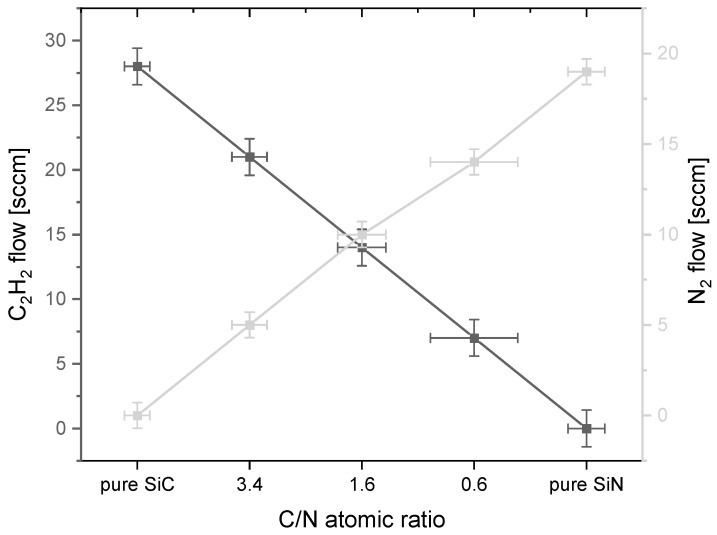
Dependence of the shares of C to N atoms on the flows of their precursors.

**Figure 3 materials-17-02450-f003:**
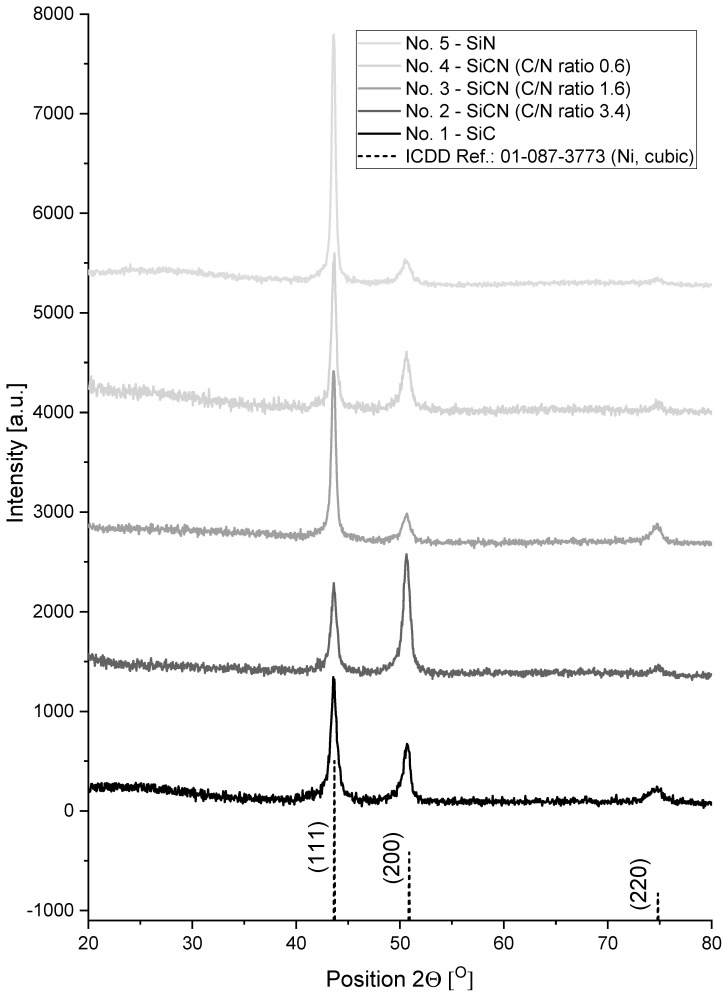
Diffractograms of coatings deposited in processes 1–5 with different C/N ratio, and pattern from ICDD card (no. 01-087-3773) corresponding to identified Ni phase.

**Figure 4 materials-17-02450-f004:**
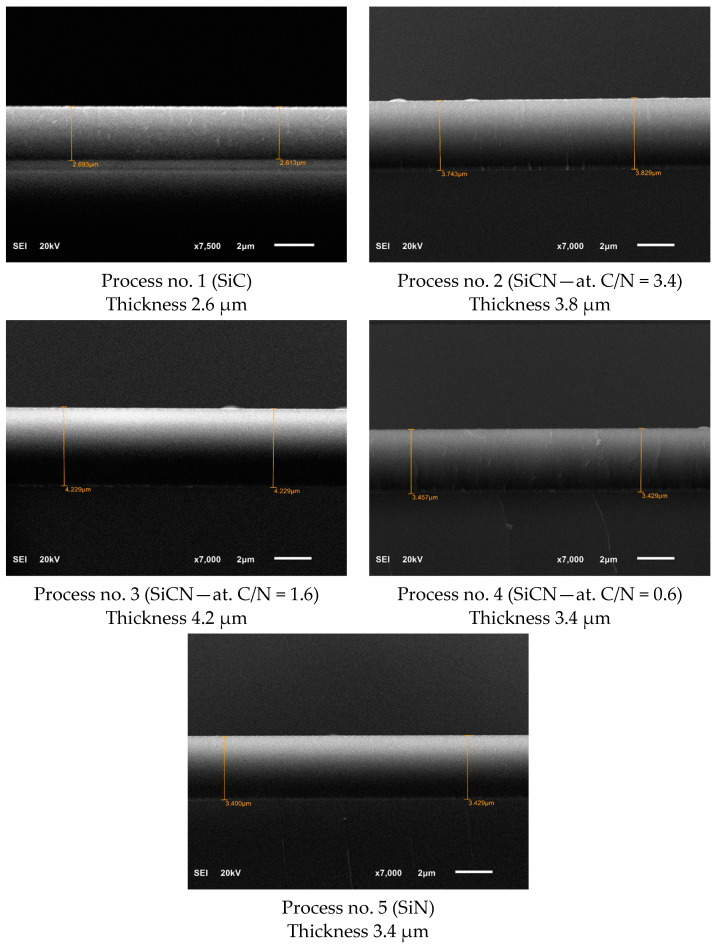
Cross-section of the coatings on a Si substrate. SEM imaging in secondary electron SE mode.

**Figure 5 materials-17-02450-f005:**
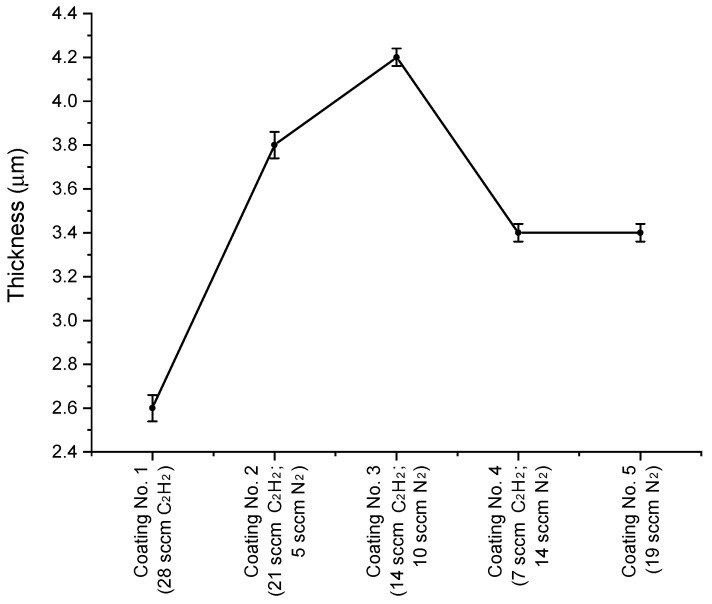
Dependence of reactive gas flow on coating thickness.

**Figure 6 materials-17-02450-f006:**
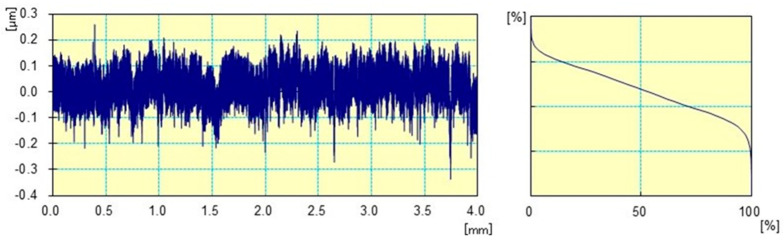
Examples of roughness measurements of 2D profiles of sample No. 0.

**Figure 7 materials-17-02450-f007:**
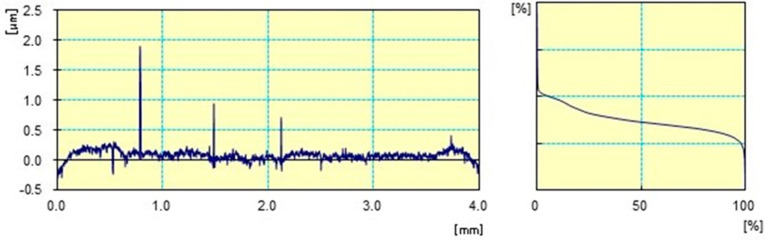
Examples of roughness measurements of 2D profiles of sample No. 1.

**Table 1 materials-17-02450-t001:** Chemical composition of tested alloy.

Element Percentage % wt.								
Cr	Mo	Si	Fe	Co	Mn	Ta	Nb	Ni
24.79	8.89	1.57	1.33	0.17	0.12	1.05	0.79	rest

**Table 2 materials-17-02450-t002:** Deposition process parameters.

Process Parameters		Coating				
		No. 1—SiC	No. 2—SiCN	No. 3—SiCN	No. 4—SiCN	No. 5—SiN
Residual pressure [Pa]		3 × 10^−3^	2.7 × 10^−3^	3 × 10^−3^	2.2 × 10^−3^	3 × 10^−3^
Etching by glow discharge plasma	Ar pressure [Pa]	1.52	1.48	1.62	1.6	1.61
	Ar flow [sccm]	25	25	17.5	17.5	17.5
	current [mA]	100	100	100	100	120
	time [min]	15	15	15	15	15
Deposition of coatings	Ar pressure [Pa]	4.5 × 10^−1^	4.7 × 10^−1^	4.7 × 10^−1^	4.5 × 10^−1^	4.5 × 10^−1^
	Ar flow [sccm]	17.5	17.5	17.5	17.5	17.5
	Bias [V]	−50	−50	−50	−50	−50
	Bias current [mA]	100	70	70	60	70
	Time [min]	240	240	240	240	240
	Power on magnetrons with Si targets [kW]	4 × 0.4	4 × 0.4	4 × 0.4	4 × 0.4	4 × 0.4
	C_2_H_2_ flow [sccm]	28	21	14	7	-
	N_2_ flow [sccm]	-	5	10	14	19
	Ar + additional gas pressure [Pa]	5.1 × 10^−1^	5.2 × 10^−1^	5.0 × 10^−1^	4.8 × 10^−1^	4.9 × 10^−1^

**Table 3 materials-17-02450-t003:** Roughness test calculation parameters.

Norma	Profil	λs	λc	n	Filtr
FREE	R	2.5 µm	0.8 mm	5	GAUSS

**Table 4 materials-17-02450-t004:** Collected results of chemical composition measurements by EDS.

Coating	Element						
	Si		N		C		at. C/N
	at. [%]	wt. [%]	at. [%]	wt. [%]	at. [%]	wt. [%]	
No. 1—SiC	24.8	38.5	-	-	75.2	61.5	-
No. 2—SiCN	29.6	46.7	15.9	12.5	54.5	40.8	3.4
No. 3—SiCN	35.2	53.3	25.2	19.0	39.6	27.7	1.6
No. 4—SiCN	42.9	61.0	35.3	25.0	21.8	14.0	0.6
No. 5—SiN	47.7	64.7	52.3	35.3	-	-	-

**Table 5 materials-17-02450-t005:** Values of water and diiodomethane contact angle.

	Substrate	No. 1—SiC	No. 2—SiCN	No. 3—SiCN	No. 4—SiCN	No. 5—SiN
Water contact angle [deg]	81.5 ± 0.6	74.7 ± 4.5	86.8 ± 2.0	76.4 ± 1.9	75.2 ± 3.2	65.4 ± 0.8
Diiodomethane contact angle [deg]	68.3 ± 0.49	54.7 ± 1.1	49.4 ± 2.6	52.2 ± 0.6	59.4 ± 0.5	54.1 ± 0.5

**Table 6 materials-17-02450-t006:** Values of surface free energy and its polar and dispersion components.

	Substrate	No. 1—SiC	No. 2—SiCN	No. 3—SiCN	No. 4—SiCN	No.5—SiN
Polar component [mJ/m^2^]	9.1 ± 0.5	9.9 ± 2.7	2.9 ± 0.4	8.1 ± 1.3	10.8 ± 2.3	16.1 ± 0.4
Dispersive component [mJ/m^2^]	18.9 ± 0.4	25.6 ± 1.2	32.0 ± 1.2	27.7 ± 0.8	22.9 ± 1.0	24.1 ± 0.2
Surface free energy [mJ/m^2^]	28.0 ± 1.0	35.6 ± 1.7	34.8 ± 1.6	35.9 ± 0.5	33.7 ± 1.3	40.2 ± 0.8

**Table 7 materials-17-02450-t007:** Surface roughness results.

Sample	Roughness Parameters				
	Ra [µm]	Rq [µm]	Rz [µm]	Rp [µm]	Rv [µm]
No. 0	0.29 ± 0.3	0.36 ± 0.3	1.93 ± 1.01	0.9 ± 0.2	1.01 ± 0.1
No. 1—SiC	0.070 ± 0.035	0.095 ± 0.03	0.73 ± 0.3	0.54 ± 0.2	0.33 ± 0.1
No. 2—SiCN	0.055 ± 0.009	0.057 ± 0.028	0.6 ± 0.21	0.42 ± 0.1	0.21 ± 0.1
No. 3—SiCN	0.066 ± 0.027	0.094 ± 0.05	0.82 ± 0.5	0.50 ± 0.2	0.31 ± 0.2
No. 4—SiCN	0.058 ± 0.06	0.077 ± 0.01	0.65 ± 0.21	0.45 ± 0.1	0.21 ± 0.07
No.5—SiN	0.044 ± 0.021	0.058 ± 0.021	0.51 ± 0.1	0.35 ± 0.1	0.16 ± 0.06

## Data Availability

The raw data supporting the conclusions of this article will be made available by the authors on request.

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
