# Peer review of "Characteristics of Si (C,N) Silicon Carbonitride Layers on the Surface of Ni–Cr Alloys Used in Dental Prosthetics"

_materials, 2024, doi:10.3390/ma17102450_

Round 1
Reviewer 1 Report
Comments and Suggestions for Authors
Abstract.
This section is correct and show the summary of the paper.
Introduction.
This investigation is a paper that presents information for researchers in the field of materials in dental prosthetics. Metals and their alloys are materials that have been used in dental prosthodontics Theses alloys that contain chromium, nickel, cobalt in their composition, can cause allergic reactions. In order to reduce corrosion and thus the risk of adverse body reactions, various modifications are applied to the surface of parts made of these alloys.
The aim of the study is to characterize silicon carbide coatings, deposited by magnetron sputtering method on the surface of nickel-chromium alloys used in dental prosthetics.
Materials and methods.
This experimental study was designed to analyze the different disks made of the Ni-Cr alloy Heraenium NA,8 mm in diameter and 10 mm high in a same experimental protocol.
This section show several subsections of the methodology (chemical composition, phase composition, cross-section of coatings and their thicknesses, wettability surface free energy, and surface roughness). The authors should explain whether the methodology is original or based on other experimental studies. Each subsection must be complemented with sufficient references for future researchers.
Results and Discussion
In the section of Results the authors included a extensive information about each subsection of materials and methods.
However, the authors authors reported a limited discussion of the results, through their analysis, comparison and limitation with other recent international studies. In fact the aspects of discussion must be expanded.
It would be more convenient if the results and discussion sections were separated in the paper.
Conclusions. This section is a repetition of methodology and results of the experimental study. The authors only must to report the most aspects of the study.
Author Response
Dear Reviewer,
thank you very much for your comments.
We have corrected the publication.
Best regards

Reviewer 2 Report
Comments and Suggestions for Authors
In this paper, the Authors study the characteristics of silicon carbo-nitride layers on the surface of Ni–Cr alloys as dental prosthetics. The topic is dealing with a really pushing and up-to date issue as to find a proper material for prosthetic dentistry. The manuscript itself is relatively well organized and clear, however, the novelty of the paper is not clearly stated.
My main concerns about this paper as follows:
- The Authors must elaborate even more on the added values and new achievements of the presented experiments.
- Figure 2-6 could be graphed together in one Figure. The characteristic peaks are not marked clearly (phase, Bragg’s reflection, Miller indices etc) and the evaluation of the XRD measurements are also insufficient. The explanation why the Si, SiN and SiC, SiCN phases are cannot be detected, while the EDS measurements supposedly revealed them.
- The informative EDS spectra are missing to support the Table 4.
- In sub-section 3.3, the stability and adherence of coating should be also discussed. Similarly, Figure 7-11 could be merged into one Figure and revise the Figure caption and text accordingly.
- Including corrosion measurements on the layers and the substrate material would improve the quality of the paper.
- In the conclusion part, the Authors should summarize the results and their novelty in a more detailed way.
- The reference list does not reflect precisely the current state-of-the art in this particular field. So, more relevant references should be provided. The citations are mostly outdated.
Comments on the Quality of English LanguageExtensive English Language ceck is necessary, there are numerous misspells.
Author Response

(The authors gave the same response as above.)

Reviewer 3 Report
Comments and Suggestions for Authors
1. Introduction: You mention that extensive use of Co and Cr based alloys in dentistry. I would recommend that you also included changes in EU legislations about these alloys (e.g. https://doi.org/10.3390/cryst10121151) to indicate importance of modify these alloys.
2. Introduction: Provides a clear background of existing literature. However, the added value and novelty of this work should be better highlighted. Only a short description of the experiments performed is given at the end of the introduction.
3. Materials and methods: Roughness of disks prior to deposition should be mentioned.
4. Materials and methods: How many samples dd you prepare per condition?
5. Materials and methods: Repeatability of measurements should be included in the text. Please include for each set of experiments that you performed the number of measurements per sample and number of samples evaluated.
6. Results and discussion: Please add error bars in Figure 1 to indicate spread of experimental values. This is important to see how significant these differences are.
7. Results and discussion: In Figures 2-6, phases and their crystallographic orientations should be added on the images.
8. Results and discussion: In your XRD patterns did you observe any shifts and/or broadening of the peaks?
9. Results and discussion: To my point of view, a broadened peak (indicating amorphous structure) appears in Figure 6 at around 20° and not in the range of 30°-40°. In the other XRD patterns this broadening is not that clear. Maybe indicate on the figure the broadened peak or use software to remove background noise. Also, in our discussion you should consider the thickness of your coatings in relevance to the penetration depth of the method (e.g. https://doi.org/10.1017/S0885715623000052)
10. Results and discussion: The discussion on the influence of deposition parameters o the microstructure and growth of the coatings is rather limited. There are several articles that you could use to strengthen the discussion. Otherwise, this works seems more like a technical report.
11. In the Abstract and introduction there is a strong emphasis given on corrosion and protective aspects of these coatings. However, after reading the manuscript there is no direct link. I would suggest that you either removed these statements or included a short discussion to support them.
12. In some parts decimals are indicated by (.) and in other parts in (,).
13. There are several minor grammatical errors.
Comments on the Quality of English Language
There are several minor grammatical errors
Author Response

(The authors gave the same response as above.)

Round 2
Reviewer 1 Report
Comments and Suggestions for Authors
In Materials and methods, each subsection must be complemented with sufficient references for future researchers.
In each paragraphe of Discussion, the authors must be complemented with sufficient references for future researchers.
Author Response
Dear Reviewer,
thank you very much for your comments. References has been added.
All corrections are highlighted red in the text.
Best regards
Reviewer 2 Report
Comments and Suggestions for Authors
The quality of manuscript has improved a lot after the revision. The answers given to the questions and comments regarding the measurements and their evaluations are correct and satisfying.
Comments on the Quality of English LanguageThe language style is fine.
Author Response
Dear Reviewer,
Thank you for your positive opinion.
Best regards
Reviewer 3 Report
Comments and Suggestions for Authors
Dear authors,
After reading the updated version of the article and your point-by-point reply to my comments, I now believe that this article is appropriate for publication.
Kind regards
Comments on the Quality of English LanguageMinor editing
Author Response
Dear Reviewer,
We have made language corrections. Thank you for your positive opinion.
Best regards